# High Ampacity On-Chip Wires Implemented by Aligned Carbon Nanotube-Cu Composite

**DOI:** 10.3390/ma16031131

**Published:** 2023-01-28

**Authors:** Xiaojia Luo, Xiao Liang, Yang Wei, Ligan Hou, Ru Li, Dandan Liu, Mo Li, Shuyu Zhou

**Affiliations:** 1Microsystem & Terahertz Research Center, China Academy of Engineering Physics, Chengdu 610200, China; 2Institute of Electronic Engineering, China Academy of Engineering Physics, Mianyang 621999, China; 3Information Materials and Device Applications Key Laboratory of Sichuan Provincial Universities, Chengdu University of Information Technology, Chengdu 610225, China; 4School of Electronic Science and Engineering, University of Electronic Science and Technology of China, Chengdu 611731, China; 5State Key Laboratory of Low-Dimensional Quantum Physics, Department of Physics and Tsinghua-Foxconn Nanotechnology Research Center, Tsinghua University, Beijing 100084, China; 6Advanced Materials Division, Suzhou Institute of Nano-Tech and Nano-Bionics, Chinese Academy of Sciences, Suzhou 215123, China; 7Key Laboratory for Quantum Optics, Shanghai Institute of Optics and Fine Mechanics, Chinese Academy of Sciences, Shanghai 201800, China

**Keywords:** super aligned carbon nanotube, CNT–Cu, atom chip, high ampacity

## Abstract

With the size of electronic devices shrinking to the nanometer scale, it is of great importance to develope new wire materials with higher current carrying capacity than traditional materials such as gold (Au) and copper (Cu). This is urgently needed for more efficient, compact and functional integrated chips and microsystems. To meet the needs of an atom chip, here we report a new solution by introducing super-aligned carbon nanotubes (SACNTs) into Cu thin films. The microwires exhibit an ultra-high current carrying capacity beyond the limit of the traditional Cu wires, reaching (1.7~2.6) × 10^7^ A·cm^−2^. The first-principles calculation is used to obtain the band structural characteristics of the CNT–Cu composite material, and the principle of its I–V characteristic curve is analyzed. Driven by the bias voltage, a large number of carriers are injected into the CNT layer from Cu by the strong tunneling effect. Moreover, a variety of microwires can be designed and fabricated on demand for high compatibility with conventional microelectronics technology. The composite structures have great potential in high-power electronic devices, high-performance on-chip interconnecting, as well as other applications that have long-term high-current demands, in addition to atom chips.

## 1. Introduction

The waveguide atomic gyroscope, which is considered a promising core element of the next-generation inertial navigation system, is expected to improve the accuracy of angular velocity measurements by several orders of magnitude [1,2,3,4]. The atom chip is the core component of the cold atomic system to control atoms. In a vacuum glass cell, the cold atoms can be manipulated by laser and magnetic field which is formed by loading current on the wires of the atom chip. A smooth and low noise single-mode atomic magnetic waveguide on an atom chip is one of the most feasible methods to realize a miniaturized waveguide gyroscope [5,6,7]. To build single-mode waveguides, a two-dimensional bound frequency of at least a few hundred kHz is necessary, which requires several amperes of current to be passed through the micrometer-wide wires [8]. Commonly used metallic materials, such as Au and Cu, do not have sufficient current-carrying capacity to withstand such high current densities [9]. Although the current carrying capacity of CNTs can be higher than 10^9^ A·cm^−2^, their electrical conductivity is low (~10^2^ S·cm^−1^), which means if CNTs are used as microwires directly, the power dissipation of current through them may damage the chip.

In recent years, CNTs exhibit excellent electrical properties after being prepared as composite materials due to their unique electricity and mechanics properties [10,11,12]. Especially, composites based on CNTs and Cu [13,14,15] have attracted the interest of researchers as a new generation of conductor materials. Theoretically, CNT–Cu composites have been confirmed to have better electrical properties than Cu [16]. And they could even reach 10^11^ A·cm^−1^ by doping with other metals [17]. Experimentally, the CNT–Cu composite made by Kenji Hata et al. exhibits comparable conductivity (2.3~4.7 × 10^5^ S·cm^−1^) as Cu (5.8 × 10^5^ S·cm^−1^) but with a 100-times higher ampacity (6 × 10^8^ A·cm^−2^) [14]. Therefore, it is possible to use CNT–Cu to form micro-wires on-chip to build the required single-mode atomic waveguides.

With the consideration of CNT–Cu compatibility with chip processing, in this paper, SACNTs film is proposed to prepare the CNT–Cu composite on chip for the first time. The preparation of SACNT is consistent with other papers [18,19]. This method is different from the traditional CNT–Cu composites [20,21]. The arrangement of CNT and the Cu coating on CNT are high-precision operations controlled on the nanometer scale. Because of the controllable orientation, the stray current in the wire can be highly suppressed [20,21,22]. The CNT–Cu composite material is obtained by means of the layer-by-layer preparation of CNT and Cu thin films. The ratio of CNT and Cu in the composite is adjusted by controlling the number of SACNT layers and the thickness of the Cu film. After that, the composite is etched into micro-wires by conventional semiconductor micro-fabrication techniques. Meanwhile, the first-principles calculation is adopted to explain the principle of the composite material with improved carrying capacity. The results show that the prepared CNT–Cu composite wire has a high current carrying capacity (>10^7^ A·cm^−2^) and nonlinear conductivity characteristics, which may be caused by the doping of CNTs by Cu.

In this paper, in order to meet the demand for atom chips with high current carrying capacity, we solved the uniform coating problem of Cu on CNT through cooperation with Tsinghua University. The current carrying capacity of this wire is one order of magnitude higher than that of traditional Cu wire, and it has higher conductivity and better thermal expansion matching ability. At the same time with lower power consumption, the wire does not easily fall off the substrate due to the actual matching of the thermal expansion coefficient under the high temperature generated by ohmic heat. Secondly, CNT films prepared based on SACNT have excellent electrical anisotropy and can effectively inhibit spurious current. At the same time, the process has good compatibility with integrated processes such as photographic fabrication, metal etching, and film deposition of chips, which is an advantage compared with other CNT–Cu composites. The above contents are more beneficial for the technology provided in this paper to be applied to the characteristics of atomic chips, and these contents are explained in the text. As this technology is oriented towards atom chips, we also overcome the integration problem of the through Si via (TSV) process and high-current-carrying composite wires in order to solve the current access problem in the glass cell of the chip in an ultra-high vacuum.

## 2. Materials and Methods

In order to keep the wafer clean with a mirror surface performance, physical deposition is used instead of the traditional electroplating technology. The production process is displayed in Figure 1. 

(a)Preparation of the CNT film.

The CNTs drawn from SACNT arrays [18,19], which are bonded with the substrate by van der Waals’ (vdW) force, are arranged parallel to the substrate. In this paper, the CNT film obtained in one such operation is called a single-layer CNT film. The filling density of CNTs in the composite can be effectively controlled through multi-layer preparation of the CNT film with good uniformity and consistency. The CNT film is first placed above the sample after being drawn from the arrays. To stick the film on a sample surface, the substrate is raised slowly to contact the film. Then the film beyond the substrate is cut by laser. A single-layer CNT film is prepared as a 2.5 cm × 2.5 cm square to measure its characteristics. 

The atomic force microscope (AFM) measurement results of the single-layer CNT drawn from SACNT arrays were conducted as shown in Figure 2. It can be assumed that the cross-sectional area of a single-layer CNT array with a width of 50 μm is approximately 2.86 μm^2^.

The volume ratio of a single-layer CNT film to a 100 nm thick Cu film is ~57%. However, the AFM result cannot accurately reflect the cross-sectional area of the array because of the overlap of the CNTs. The diameter of a single CNT is about 20 nm, based on the transmission electron microscope (TEM) results shown in Figure 3b. In general, along the axial direction of the CNTs, the calculated resistance is ~1.13 kΩ, resistivity is ~6.46 × 10^−6^ Ω·cm, and conductivity is ~1.55 × 10^5^ S·cm^−1^. Though the conductivity of the CNT film is less than that of Cu material, which is about 10^6^ S·cm^−1^, it has a high resistance anisotropy ratio of 96.5 between the axial direction and the radial direction. That means the CNT film drawn by SACNT arrays is more conducive to the transport of carriers in a specific direction, which is beneficial to reduce the current noise [21].

(b)Fabrication of the CNT–Cu composite. 

The structure of the CNT–Cu composite is Si/SiO_2_/CNT/Cu/CNT/Cu as shown in Figure 1a. First, 500 nm thick SiO_2_ is prepared by plasma-enhanced chemical vapor deposition (PECVD) on a Si substrate as the insulation layer. The CNT film is densified by atomizing ethanol to improve the contact with the substrate after being prepared on the sample. Cu film is deposited on the CNT film by electron beam evaporation (EBE) with a growth rate of 0.2 nm·s^−1^. 

(c)Fabrication of the micro-wires. 

The shape of the micro-wire is prepared using ultraviolet lithography technology with AZ5214 photoresist. The CNT–Cu composite materials are then etched layer by layer. Although the ion beam etching (IBE) can etch both CNT and Cu with oxygen, it also etches the AZ5214 at a high speed. To ensure the wires are undamaged, Cu layers are etched by IBE with an etching voltage of 350 eV with argon, and CNT layers are etched by inductively coupled plasma (ICP) at 300 W of power in oxygen. The etching time depends on whether the Cu or CNT is etched completely. An annealing treatment of 250 °C for 3 h in H_2_ gas with a controlled flow of 150 sccm as the reductive gas is carried out after the wires are formed. 

Distinguishing from the sample used for the electrical test, the ultimate atom chip is shown in Figure 1c. In order to realize the packaging of the atomic chip in a vacuum system, we combined the CNT–Cu composite wire with the TSV structure chip process to prepare the sample shown in Figure 1c.

## 3. Results

The images of the initial stage of the Cu growth on CNT film using EBE are shown in Figure 3. Cu atoms are adsorbed on the surface of CNTs by vdW forces forming crystal grains. Thus, Cu crystal grains gradually grow and accumulate to coat CNTs. This proves that our process is sufficient to complete the coating of CNTs with Cu.

We prepared various CNT n/Cu m nm ×2 samples (n stands for the number of CNT film layers, *n* = 3, 5, 7, 9; m stands for the thickness of the Cu film, m = 100, 200, 300; ×2 means CNT and Cu are prepared twice successively.) to test their electrical features with the probe station. Since our purpose is to measure the maximal current carrying capacity of the wire, the I–V curves are measured with two probes. The conductive silver glue is used to improve the contact between the probe and the sample. The test results of a short distance on a copper plate under this method show that the contact resistance of the device is less than 0.1 Ω. The I–V curves of the samples with relatively good current carrying features are plotted in Figure 4a. The common feature of these samples is that m:n is about 33. The preliminary experimental results show the CNT 3/Cu 100 nm ×2 sample has the maximum current carrying capacity, which may be determined by the uniformity and proportion of the composite.

We focus on the CNT 3/Cu 100 nm ×2 sample and try to improve its current carrying capacity. An annealing treatment is performed after the wire preparation is completed. Finally, CNT–Cu composite wires with 1 cm length and 100 μm width are prepared. The SEM photo of the processed sample surface is shown in Figure 5a. The surface structure of the sample is dense with fine Cu crystal particles. Three obvious single-layer cross-sections in the sample in Figure 5b are taken to measure the thickness of the interface, which are 170 nm, 214 nm, and 142 nm, respectively. Considering the 45° angle observation used in the test simultaneously, the sample thickness is recorded as 248 nm ± 54 nm. The measured I–V curve of the sample is shown in Figure 4b,c. The voltage rises from 0 V to 40 V in 5 s, with the maximum current set to 5 A, for protecting the sample and test system. The composite wire current reaches the current limit value of 5 A, when the voltage is 12 V. In addition to the current carrying capacity of composite wire being significantly higher than that of Cu wire, the conductivity characteristics are also different in the initial stage. For Cu wire, the current rises with the increasing voltage linearly in the initial stage, and then the growth rate slows down and tends to be saturated. However, the current growth rate of the composite wire increases with the voltage untilthe current limit of the source is reached.

Based on the above test results, the current carrying capacity of the sample is (1.7 ~ 2.6) × 10^7^ A·cm^−2^, and the conductivity is (1.38 ~ 2.14) × 10^6^ S·cm^−1^, which is much higher than 1.55 × 10^5^ S·cm^−1^ of the pure CNT sample and 2.7 × 10^5^ S·cm^−1^ of the prepared pure Cu wire on the same substrate. The I–V curve of Cu wire rises linearly at first, and gradually approaches saturation. Finally, the Cu wire is broken. The I–V curve of the CNT–Cu composite rises from non-linearity to the upper limit of the current source current. During the test, a high temperature caused by ohmic heat is generated, which is maintained for ~2.5 s before fusing. The current carrying time of the composite wire has met the requirements of the cold atom experiment, while the conventional period of an atomic interference experiment is within 1 s [1]. 

At the same time, compared with pure CNT, its adhesion on the substrate is stable, and it is not easy to cause the wire to fall off due to the thermal expansion coefficient mismatch at high temperatures. The comparison result is shown in Figure 6a,b.

## 4. Discussion

### 4.1. Magnetic Trap Analysis

To show the advantages of CNT–Cu wire in atom chips, the magnetic traps of atom chips with traditional Cu wire and CNT–Cu wires are calculated by COMSOL based on the above research results. The model is a classical 3-lines structure. The cross-section of the designed wires is 40 μm × 10 μm with a distance of 60 μm between them. An infinite meta-field is added to the outer layer of the model. An extremely refined free tetrahedral mesh controlled by a physical field is adopted. Usually, a single Cu wire is designed to carry 2 A in maximum to avoid wires broken for thermal expansion. According to the experimental results of this work, we assume that the current carrying capacity of the composite wire is 120 A. The simulation results are shown in Figure 7. We can conclude from the simulation results that the magnetic trap gradient is positively correlated with the current, and the magnetic trap gradient of CNT–Cu reaches 60 times Cu. 

### 4.2. First Principles Calculation

The conductive properties of CNT–Cu composites with electric fields parallel to carbon tubes have been discussed in detail [16]. In this work, the electrodes actually only touch the Cu layer. The outer Cu layer is at a high potential and the inner CNT is at a low potential. To analyze the phenomenon of CNT–Cu composite wires with high ampacity, the first-principle calculation is used. According to the experimental results above, we make an approximation that the CNT is completely wrapped by Cu, so that the contact plane between them can be regarded as the infinite plane between Cu and graphene, as shown in Figure 8. 

The calculation method we applied in this work is the first-principles calculation method based on the density functional theory (DFT) with vdW correlations. The vdW correction method used in our study is the vdW-DF2 formula [23] with the GGA exchange function proposed by A. D. Becke (B86b) [24]. This combination of vdW and exchange functions is suggested by I. Hamada, named revB86b-vdW-DF2 [25]. In most previous studies, this method is well performed for layered system calculations, especially for 2D materials [25,26,27,28]. We employ the framework of the projector augmented wave (PAW) method [29,30]. The cut-off energy for the plane-wave basis was set to 500 eV. For a sampling of the Brillouin zone, the Monkhorst–Pack scheme [31] and 22 × 22 × 1 k-point grid are used for calculations. To model the graphene/Cu interfaces, we constructed a superlattice structure with a slab of 10 layers Cu (111) of about 18.8 angstroms thick along with the graphene on top, and the lattice mismatch is only 3.5%. To avoid the interaction between the two cells, a 15-angstromangstrom-thick vacuum layer was inserted, as shown in Figure 9a.

The band structure of the graphene/Cu(111) system is shown in Figure 9b, the red and blue lines represent the component of C and Cu atoms, respectively. Here, the Fermi level is defined as 0 eV, and the system generally exhibits metallic properties. It should be noted that the key feature of the Dirac cone in graphene at the ”K” point along high symmetry lines is reserved, which means the great conductivity and topological properties of Dirac fermions in graphene are little affected by Cu substrate. The Dirac cone is shifted below the fermi level to 0.526 eV, which indicates the N doping of graphene due to electron injection from Cu film. In addition, one can see there are some electron states hybridization near the Fermi level, which indicates the strong exchange effect of these two materials. Therefore, during the transport processes, the electron of the surface in Cu layers can transfer much faster and topologically protected from backscattering due to the electron tunneling and states hybridization, which may be the origin of the increasing current carrying capacity in this system. Figure 9c shows the projected density of the state of Cu d orbitals and C p orbitals. It is found that the states near the Fermi level in the graphene layer are much fewer than that in the Cu layers, which means the graphene layer itself contributes little to current carrying capacity, and the electron tunneling and states hybridization at the graphene/Cu (111) interface should be the key factor. To evaluate the tunnel barrier height of the two junctions, we calculated the effective potential ($V_{\mathrm{eff}}$) distribution along the normal direction in the graphene/Cu heterostructure. The effective potential is defined as as $V_{\mathrm{eff}}\left(n\right)=V_{H}\left(n\right)+V_{\mathrm{xc}}\left(n\right)+V_{\mathrm{ext}}$, where $V_{H}\left(n\right)$, $V_{\mathrm{xc}}\left(n\right)$, and $V_{\mathrm{ext}}$ are the Hartree potential, exchange–correlation potential, and potential energy from other electrostatic interactions, respectively. The tunnel barrier $\Phi_{\mathrm{TB}}$ denoting the electron tunnel barrier height at the junction interface defined by $\Phi_{\mathrm{TB}}=\Phi_{\mathrm{gap}}-\Phi_{\mathrm{Cu}}\;$can therefore be calculated from the effective potential envelopes, where $\Phi_{\mathrm{Cu}}$ stands for the $V_{\mathrm{eff}}$ of the Cu–Cu bond potential energy and $\Phi_{\mathrm{gap}}$ is the interface gap potential energy. The calculated $\Phi_{\mathrm{TB}}$ of graphene/Cu is –8.09624 eV. 

According to the calculation results, the tunnel barrier decreasing with the external electric field increasing causes the measured nonlinear I–V curve. With the strong tunneling effect, a big number of carries inject from the Cu layer to the CNT layer, and transport like Dirac fermions due to states hybridization, and it causes the current to increase rapidly and saturation at a voltage equal to $\Phi_{\mathrm{TB}}$, which agrees with our experiment results. At the same time, another possible reason for the increased current-carrying is the high mechanical strength of the carbon nanotubes makes the composite wire harder to break under electronic impact [32].

## 5. Conclusions

In this work, we developed a new method to prepare a CNT–Cu composite based on the SACNT. The wires of 1 cm length and 100 μm width reach a high current capacity (1.7~2.6) × 10^7^ A·cm^−2^ and a maximum conductivity (1.38~2.14) × 10^6^ S·cm^−1^ in the measured range which is better than Cu film wires. Meanwhile, this technology is easy to integrate on Si-based chips or other substrates with good insulation. Without considering the complexity of the process, the composite wire could have better uniformity and performance with the circulation of 1 layer SACNT and a 33 nm Cu film. The thickness of the wire can be controlled by the number of cycles. It can be used in devices requiring high current with limited volume, not only in atom chips. The first-principles calculation results explain the high current carrying capacity of composite materials and the reasons for the nonlinearity of the I–V curve. When the voltage reaches a certain value, some carriers begin to transport on the CNTs, resulting in the current-carrying characteristics of the composite material. This also provides theoretical guidance for the further optimization of composite materials.

## Figures and Tables

**Figure 1 materials-16-01131-f001:**
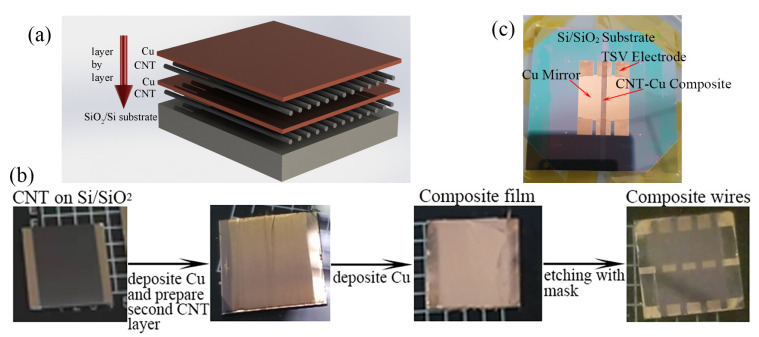
(**a**) Schematic diagram of the CNT-Cu composite preparation process; (**b**) technological process; (**c**) The photo of the ultimate atom chip with CNT–Cu wires.

**Figure 2 materials-16-01131-f002:**
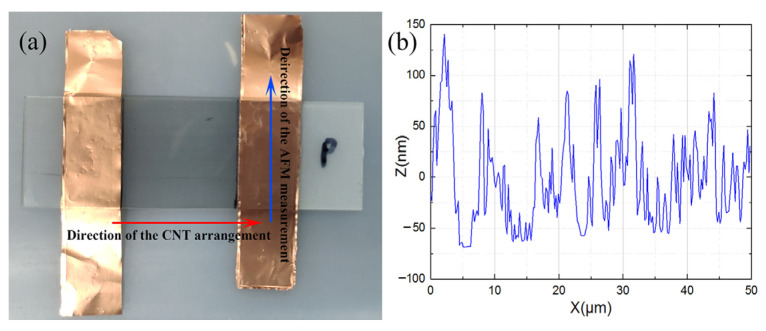
(**a**) The prepared single-layer CNT film without being densified; (**b**) Measured AFM result of the CNT film.

**Figure 3 materials-16-01131-f003:**
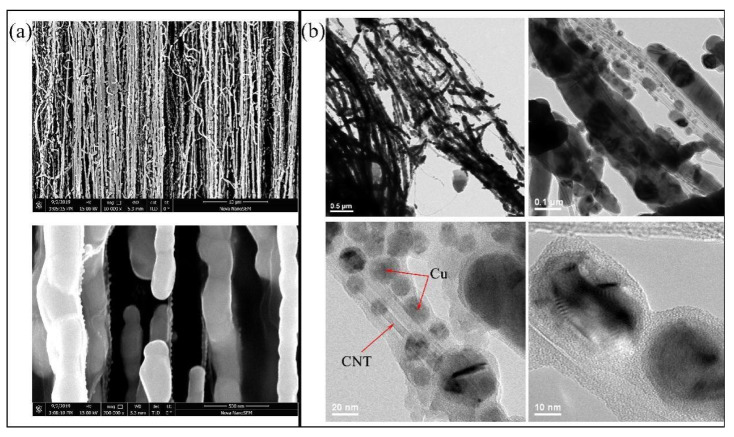
The images of Cu growth on SACNT at the initial stage, (**a**) scanning electron microscope (SEM) and (**b**) TEM.

**Figure 4 materials-16-01131-f004:**
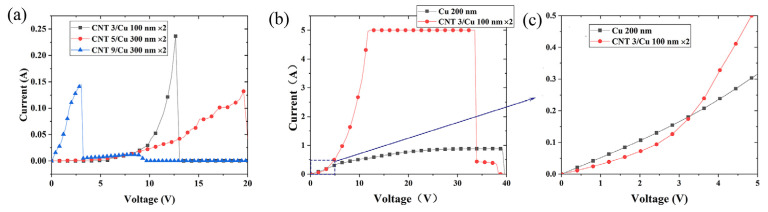
The meaning of the label, e.g., CNT 3/Cu 100 nm ×2 means the sample is made up of three layers of SACNT film and 100 nm Cu film with two cycles. (**a**) I–V curves of the various layers CNT–Cu composites with higher current-carrying samples. (**b**) Comparison of I–V curves between pure Cu wire and CNT–Cu composite wire, and enlargement in the dotted red box (**c**).

**Figure 5 materials-16-01131-f005:**
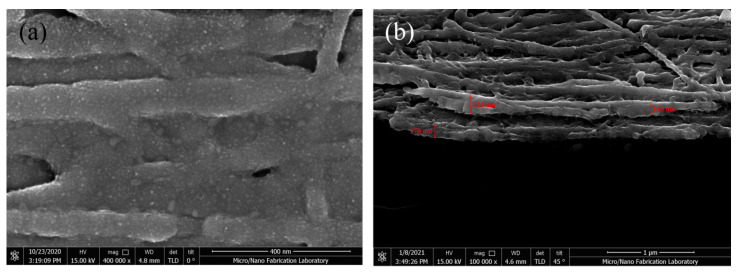
The SEM photos of the CNT-Cu composite cross-section viewed at 90° and (**a**) at 45° (**b**).

**Figure 6 materials-16-01131-f006:**
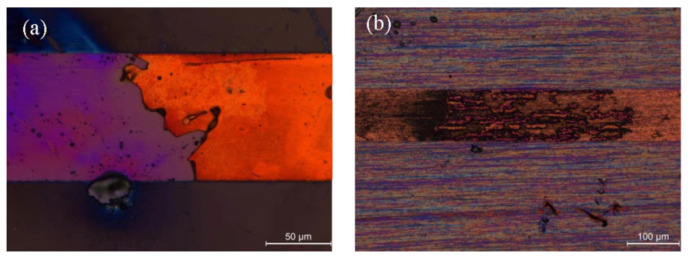
The optical microscope photos of wires after current-carrying, (**a**) The shedding of pure Cu wire caused by high temperature, (**b**) CNT–Cu composite wire after burning.

**Figure 7 materials-16-01131-f007:**
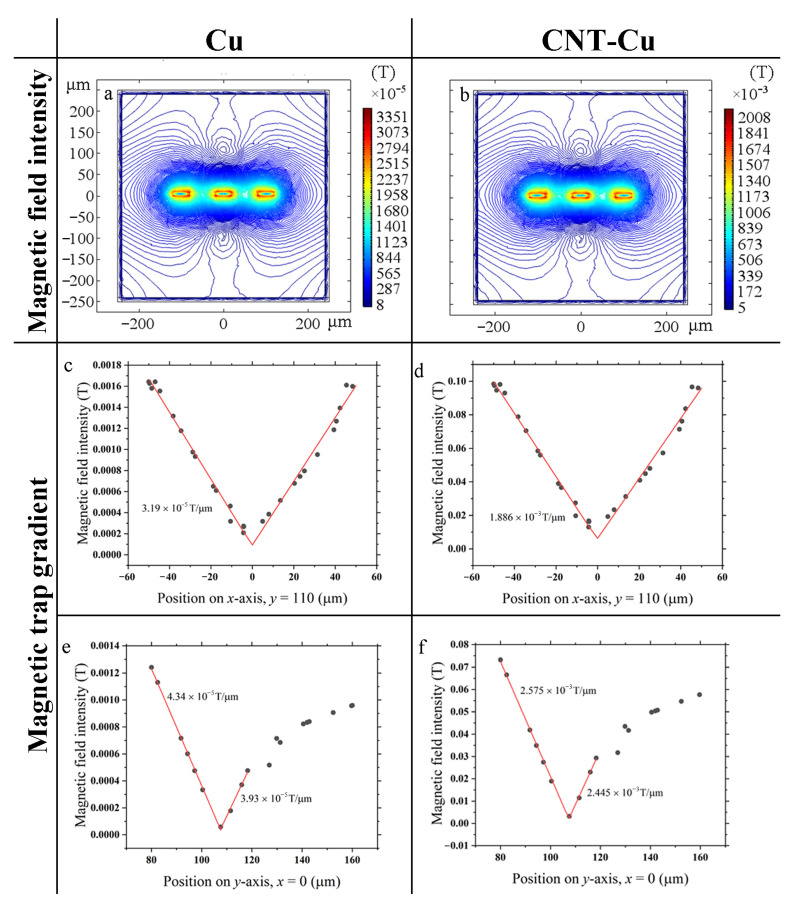
The calculation results of the magnetic trap produced by Cu wires and CNT–Cu wires. The black dots are the finite element simulation results. The red lines are the fitting date to calculate the magnetic trap gradient. The magnetic trap is produced by Cu wires (**a**) and by CNT–Cu wires (**b**). (**c**,**e**) is the data of the cutting line crossing the magnetic trap in (**a**) along the x-axis and y-axis, respectively, while (**d**), and (**f**) is for (**b**).

**Figure 8 materials-16-01131-f008:**
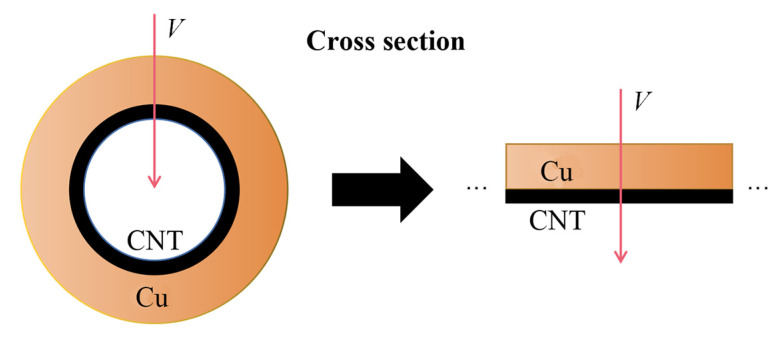
The structure model diagram of the CNT–Cu composite.

**Figure 9 materials-16-01131-f009:**
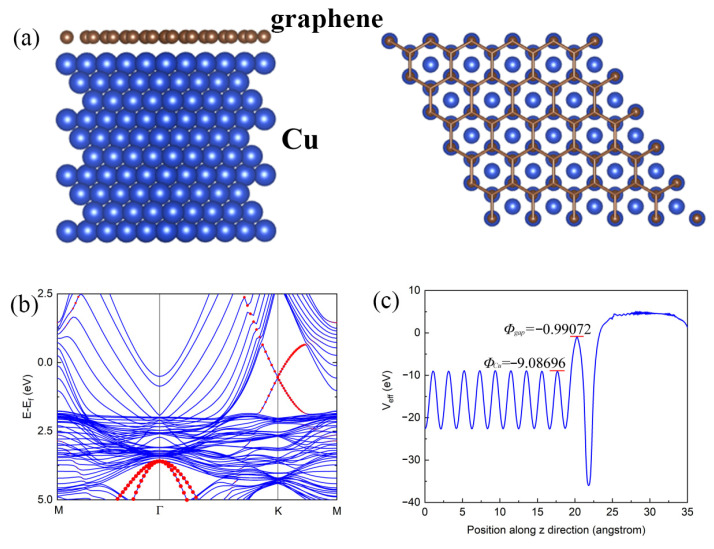
(**a**) Model structure; (**b**) Band structure; (**c**) Average local potential.

## Data Availability

Not applicable.

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
