# Peer review of "High Ampacity On-Chip Wires Implemented by Aligned Carbon Nanotube-Cu Composite"

_materials, 2023, doi:10.3390/ma16031131_

Round 1
Reviewer 1 Report
Journal: Materials (ISSN 1996-1944)
Manuscript ID: Materials- 2158232
The authors presented an article on “High ampacity on-chip wires implemented by aligned carbon nanotube-Cu composite”. I think the article is well organized and suitable for the "Materials" journal. However, the article will be ready for publication after a major revision. Comments are listed below.
1. A sentence about numerical results should be added to the abstract part.
2. The introduction section is insufficient. It should be further explained, supported by the literature.
3. In the fabrication of CNT-Cu composites; Argon was used during etching, and hydrogen was used in annealing treatment. Why is that?
4. Some shape labels are too long. It can be shortened.
5. It can be discussed by giving examples from the literature in the Results and discussion section.
6. The inscriptions in Figure 1c are not legible.
7. Units are not spelled correctly. The entire article should be reviewed.
8. The article contains numerous typographic and language errors. It should be corrected.
9. The article should be rearranged by taking into account the journal writing rules and citation rules.
10. The article is well-organized, yet there is a reference problem. First, your reference list contains no article from the “Materials” journal. If your work is convenient for this journal's context, then there are many references from this journal. Secondly, cited sources should be primary ones. Namely, the indexed area shows the power of a paper and directly your paper's reliability. Please make regulations in this direction.
Turnitin similarity rate is 17%.

Reviewer 2 Report
This manuscript presents super-aligned carbon nanotubes (CNTs) - Cu composite for chip microwires application. Authors show that fabricated microwires have current carrying capacity beyond the limit of the traditional Cu wires.
My main concern is the novelty of presented results and the methodology used to explain underlaying pheneomena.
1) It was already shown by many authors [10.1038/ncomms3202; 10.1039/c5nr03762j; 10.1088/0957-4484/27/33/335705; 10.1098/rsos.180814 ] that wires composed of alligned CNT combined with Cu can succesfully rival purely Cu wires.
2)Authors claim that their ab-initio calculations allowed to explain the mechanism of improved electrical performance of the composite. But unfortunately, this study does not provides any insights in understanding CNT-Cu composites.
(i) The structural, electronic and the charge transport properties of Cu-CNT composite were already fully explained in the following works: 10.1039/c5cp01470k; 10.1039/C7NR02142A; 10.1039/C8NR07521B.
(ii) The considered model of Cu-CNT composite shown in Fig. 9 is not a model of Cu-CNT composite but rather graphene-Cu interface. Graphene-Cu interface is very different from full Cu-CNT composite model that should be considered. For instance authors do not consider structural changes in CNT structures induced by Cu matrix or differnt types of CNTs as well as the impact of the concentration of CNT content on the composite properties. Authors should repeat their DFT calculations taking into account CNT instead of graphene. Results should be compared to existing literature data: 10.1039/c5cp01470k; 10.1039/C7NR02142A.
Other comments:
3)The existing breakthroughs in the preparation of Cu-CNT composites as well as existing theoretical works explaining good performance of Cu-CNT composites are not sufficiently discussed in the introduction.
4)What kind of nanotubes were used in the preparation of composites?
5)What is the concentration of the defects, especially oxygen bearing defects in these CNTs? It would be to show some Raman analysis.
6) What was the length of these CNTs?
7) What was the concentration of CNTs in Cu-CNT composites. How the charge transport properties of composite changed with changing concentration of CNTs in the composite? How the authors finding agree with literature?
8) I would like to see the quality of the interface between CNT and Cu and between CNT and SiO2. Additional EDX analysis is needed.
9)For the sake of reproducibility, authors should provide all the parameters used for COMSOL simulations.
Therefore, I do not recommend this manuscript for publication in Materials in the present form.
Author Response
Please see the attachment page 4-6.

Round 2
Reviewer 1 Report
The authors made the desired corrections. I believe this article can be accepted for publication in the "Materials" journal in its final form.

Author Response
Thank you very much for your approval.
Reviewer 2 Report
The manuscript has been revised, however, model used to support experimental results is still completely incorrect. Instead of considering CNT-Cu interface authors consider graphene-Cu interface. Authors also did not correct the model description in the text and in the Fig. 9. They refer to graphene monolayer as to carbon nanotube on p.8 and p.9.
Even though experimental results are interesting, the modelling part is incorrect and therefore, I do not recommend this manuscript for publication in Materials in the present form.
Author Response
To avoid ambiguity, we have changed the CNT to graphne on p. 8 and p. 9. Limited by the calculation conditions, after discussion, we considered we have made an acceptable approximation. It is difficult for us to establish a model with high external potential and low internal potential for Cu to completely surround CNT, and the calculation amount is large. And, the calculation results can reflect why the resistance decreases with the increase of voltage in the experiment. We hope you can accommodate the concessions we made in the calculation.
Round 3
Reviewer 2 Report
I am satisfied with provided explanations and corrections. For clarity, the description of the model should be also changed in Fig. 9 a. Now graphene is marked as CNT on the atomistic model visualisation. I recommend this manuscript for publication in Materials for publication after minor revision.
Author Response
Thank you for your kindness. We have change Figure 9 and fixed the spelling error. The font in fiugre 8 is also been fixed.